# Antecedents of Consumer Food Waste Reduction Behavior: Psychological and Financial Concerns through the Lens of the Theory of Interpersonal Behavior

**DOI:** 10.3390/ijerph182312457

**Published:** 2021-11-26

**Authors:** Saman Attiq, Amanda M. Y. Chu, Rauf I. Azam, Wing-Keung Wong, Sumia Mumtaz

**Affiliations:** 1Air University School of Management, Air University Islamabad, Islamabad 54000, Pakistan; saman.attiq@mail.au.edu.pk (S.A.); sumiamumtaz@gmail.com (S.M.); 2Department of Social Sciences, The Education University of Hong Kong, Tai Po, Hong Kong; amandachu@eduhk.hk; 3Punjab University of Technology Rasul, Punjab 50380, Pakistan; vc@putrasul.edu.pk; 4Department of Finance, Fintech & Blockchain Research Center, and Big Data Research Center, Asia University, Taichung City 41354, Taiwan; 5Department of Medical Research, China Medical University, Taichung City 40447, Taiwan; 6Department of Economics and Finance, The Hang Seng University of Hong Kong, Shatin, Hong Kong

**Keywords:** food waste, anticipated negative emotion, awareness of consequences, habit, financial concerns, waste reduction behavior

## Abstract

This study sought to investigate the role of consumers’ emotional, cognitive, and financial concerns in the development of food waste reduction, reuse, and recycling behavior among restaurant patrons. Food waste in restaurants is a major problem for the food service industry, and it is a growing source of concern in developing countries, where eating out is becoming increasingly popular. A large portion of restaurant food waste in these markets originates from the plates of customers, highlighting the importance of consumer behavior changes in reducing waste. The current study has used a quantitative approach to analyze the impact of anticipated negative emotion of guilt, awareness of consequences, habit, and financial concern on food waste reduction behaviors, i.e., reduce, reuse, and recycle. The study collected 492 responses and data is analyzed for hypotheses testing through Partial Least Square-Structural Equation Modelling. The findings showed that anticipated negative emotions of guilt, awareness of consequences, habit, and financial concern have a significant impact on restaurants’ consumer food waste reduction behaviors. Managers, policymakers, and researchers interested in resolving the food waste problem will find the study useful. Other topics discussed include the implications and limitations as well as possible future research directions.

## 1. Introduction

The challenge of food waste exists worldwide, and the financial, economic, social, and environmental impacts can be significant [1]. For instance, food waste causes several environmental issues such as pollution, climate change, and global warming [2,3]. When an edible item is left unconsumed, such as when supermarkets discard it because of blemishes or an item’s unwanted color, it is food waste [4]. From a social point of view, survey reports such as [5] showed that approximately 800 million people are going hungry or being malnourished. It was also found that nearly one-third of globally produced food is wasted (i.e., 1.3 billion tons/year) which causes around $750 billion yearly financial loss [6].

Food wastage is a serious threat for both developed and under-developed countries [7,8]. According to some global statistics, the amount of food that is wasted each year is approximately $1 trillion while in developing countries it is estimated at nearly $310 billion. Food waste has a number of negative consequences, including strain on natural resources such as land and water as well as a financial liability [5,9,10]. Avoidable food loss occurs at every stage of the food supply chain, necessitating the establishment of formalized systems for the management of food waste [11,12,13].

A large portion of food waste most often takes place at the consumption stage (35%) [14]. A lot of studies have been compiled for food waste in household sector [15,16], even in restaurants from a restaurant perspective [17,18]. Sirieix et al. [19] examined the role of culture, social norms, and emotions in developing restaurants consumers’ intentions. Recent studies call for further research to offer an inclusive model addressing and developing effective policies to reduce the problem of food waste. Thus, it is necessary to study consumer behavior in relation to food and, in particular, to explain the drives of consumer food waste [20]. Stancu et al. [21] and Stefan et al. [22] determined a positive connection between food waste and surplus food. Wang et al. [23] highlighted the importance of the determinants of consumer food waste reduction behavior in the food service sector. Some of the recent studies acknowledged that the growing magnitude of food wastage at restaurants undermines environmental sustainability [17]. The larger share of food waste comes from customers which requires to be examined to address the issue of food wastage [24]. Though there is an increasing trend of dining out [23], and food wastage at restaurants is frequently discussed in the media [17], still it was acknowledged that this sector is discussed less rigorously despite being recognized as a key challenge [25].

The study aimed to examine the impact of consumer’s emotional, cognitive, and financial concerns in the development of food waste reduction, reuse, and recycling behavior of restaurants’ consumers.

### Significance of the Study

A review of prior research suggested some open gaps. First, prior literature related to food waste was extensively researched by using a qualitative research approach and focused on the identification of motives and barriers to minimize food waste [26]. Recent studies attempted to examine the drivers of food waste reduction by using a quantitative research approach [17,27]. Despite scholars (e.g., [17]) recommending the need for more empirical research on better understanding food waste consumer behavior, it appears that there is not yet sufficient empirical research to adequately help understand this phenomenon. Secondly, consumers’ food waste reduction behavior has been found to be extensively investigated using the theory of planned behavior, which uses a cognitive approach to explain behavior [28,29,30,31]. Scholars suggested considering non-cognitive variables such as emotions and habits in explaining food waste behaviors (e.g., [10]) has been limited. The theory of interpersonal behavior is a suitable approach in hand to explain food waste behaviors through cognitive, emotions, and habits [32]. Third, it was found that consumer food waste reduction is a major challenge and its complexity is still unclear, as it is linked to consumer behavior [33,34,35]. Consumer behavior is complex, and a variety of unknown factors may contribute to food waste. Similarly, multiple dimensions may uncover the food waste reduction behavior such as reuse, recycle, and reduce [35]. It was found that recycling and reuse behaviors are extensively focused while reducing food waste behaviors were largely ignored [36]. The reduction is one of the most important factors in explaining the food wastage concept [37]. Thus, the use of 3Rs (reduce, reuse, and recycle) provides a more comprehensive measure in examining food waste reduction behavior as compared to traditional measures [7,37]. Fourth, a major proportion of food is wasted at the end of the food supply chain such as in the food service industry (restaurants and hotels), and few restaurants consider food waste reduction and recycling [38]. Though consumer food waste behavior has received some attention [39,40], food waste behavior in the context of restaurants and dining out is still understudied [24]. The amount of food wasted during eating in restaurants depends on consumer behavior such as ordering more than what is required for eating and leaving food on the plate [41]. Obtaining a wider insight into consumer food waste behavior in restaurants can help reduce this important source of food waste [24]. Fifth, it was found that young consumers are more likely to waste food [42], but, at the same time, young consumers are more aware of the negative consequences of food waste [43], and recognize the importance of recycling [44]. These contradictory results of young consumers regarding food waste reduction [45] require further exploration. The current study seeks to address the above-mentioned gap by developing a research model based on the theory of interpersonal behaviors and examining it by using the data of young consumers in the context of restaurants.

Additionally, it demonstrates practical significance in addition to theoretical significance. This study contributes both managerially and theoretically to understand the consumer’s viewpoint. Theoretically, the study will examine the variables quantitatively and examine the factors that directly affect young consumers’ food waste reduction, reusing, and recycling behavior. Managerially, this study will assist the food service sector, such as restaurants, hotels, and government, to plan programs to encourage food waste reduction in terms of reducing, reusing, and recycling the food instead of wasting it.

The paper is structured as follows: the study introduces the research problem, identifying literature gaps in the first part. The second part contains a literature review on variables. The third part of this study describes the methodology, data analysis, and conclusion along with the implications, limitations, and suggestions of the study.

## 2. Theoretical Model and Hypotheses Development

### 2.1. Theory of Interpersonal Behavior (TIB) as Theoretical Lens

Food waste is a significant global issue; thus, bringing consumers to participate in waste reduction practices through changed behaviors is important. However, stakeholder (i.e., food consumers) participation is required in order to adopt these sustainable initiatives. Due to the complexity of human behavior, the most common theory, the Theory of Planned Behavior (TPB) seems to be unable to explain the emotional aspect of consumer behavior. TPB explains behavior through intention and is considered as a static model, based on self-interest motives, and excludes emotional and non-conscious influence [31]. To cover this gap, the theory of interpersonal behavior (TIB) was used to support this study. It was proposed by [46], in contrast to the TPB. It documented the vital role of habit and emotions in forming intentions performing a behavior. It also says [47] that “intention is the result of personal/social norms, cognition of consequences, and emotion/affect. TIB presents justifications regarding emotions along with cognitive and social aspects to predict the behaviors of consumers [48]. In addition, it is also stated that the likelihood of executing a behavior not only is subjected to the person’s habits, but also to situations that facilitate the behavior and intentions [47,49]. During food consumption and wastage, negative emotion is an important psychological aspect in measuring food waste reduction behaviors [50,51]. The missing influence of habit and emotions in the former studies are unable to effectively measure the psychological antecedents of consumer behavior towards food waste [32]. Lack of consumer food waste awareness about consequences is affecting the environment in developing countries [52]. Consumer awareness of the consequences of food waste is considered important as it requires changing the attitude and behavior of consumers [53].

### 2.2. Hypotheses Development

#### 2.2.1. Waste Reduction Behaviors (Reuse, Reduce and Recycle)

Food waste exists in many distinct but intertwined aspects of daily life, including shopping, storage, cooking, and eating. Consumers are unaware of all of the factors that contribute to the food waste they produce, as they are so engrossed in their daily lives [54]. Improving 3Rs in the foodservice industry and among the consumers remains a primary challenge [55]. Reduce is the reduction of waste generation, Reuse is utilizing of products and parts in some form or another again, while Recycle is the use of food parts as resources for making some other useful things. Reducing and recycling food waste is becoming an important discussion point for sustainability. Recycling food waste can improve the environmental conditions and thus the limited minerals and nutrients in the soil can be reutilized efficiently [56]. The food waste behavior in this study is examined through the 3Rs (reduce, reuse and recycle), where food waste can be minimized, reused, or recycled in a manner that it can be used as a raw material for composite [57]. Few studies have examined the recycling aspect of the 3Rs, yet a detailed focus on all three is required as a consumer food waste behavior.

#### 2.2.2. Anticipated Negative Emotion and Waste Reduction Behaviors

Emotion refers to “mental and physiological feeling states (positive or negative) that are short-lived, but are usually intense” [58]. Anticipated emotions are defined as “one’s expected emotional/feeling responses that one will experience by engaging in a particular behavior in the future” [59] such as the anticipated negative emotions of guilt, regret, and remorse. When expectations are not met they generate negative emotions of sadness, anger, and disgust” [60]. Negative emotions are found to be positively and significantly related to intention to conserve the electricity and expression of sadness, anger, guilt, and embarrassment on not saving electricity [58]. It was found that Anticipated Negative Emotion (ANE) has no significant influence on behavior to reuse recycled water [60]; and is not associated with pro-environmental behavior [61]; yet has a direct effect on recycling behavior of youth [59]. ANE has significantly influenced the behavior of tourists [62]. ANE of guilt was found to positively influence young consumer’s waste reduction intention at tourist destinations [63]. According to the findings of the previous studies, consumers’ intention to reduce food waste is positively related to their anticipated negative emotions [32] as well as green behavior [61]. On the basis of the previously cited literature, it can be predicted that:

**Hypothesis 1a** **(H1a).***Anticipated negative emotion of guilt has a significant impact on consumer’s food waste reduction behavior*.

**Hypothesis 1b** **(H1b).**
*Anticipated negative emotion of guilt has a significant impact on consumer’s food waste reusing behavior.*


**Hypothesis 1c** **(H1c).***Anticipated negative emotion of guilt has a significant impact on consumer’s food waste recycling behavior*.

#### 2.2.3. Awareness of Consequences and Waste Reduction Behaviors

“Awareness of Consequences” (AOC), is described as “the cognition that an individual believes that failure to perform a specific behavior may bring adverse consequences to others” [64]. Awareness of consequences influences the beliefs that individuals’ actions can affect others if actions have not been changed [65]. The AOC indicates a consumer’s awareness concerning performing or not performing a behavior. Tourists’ awareness of consequences, impacts their personal norms and binning intention to show their pro-environmental behavior in national parks [66]. AOC has shown a significant positive relationship to attitude and internal motivation examined by [67]. AOC indirectly influences behavioral intention [68]. Likewise, when people are aware of the consequences of recycling on the environment, they like to pool in their efforts to recycle their food waste. Thus, researchers found a positive significant association between waste behaviors with recycling intention [7]. The more consumers have the awareness of the environmental consequences, the higher their behavioral intention [69]. Awareness of food waste consequences are related to sub-optimal food, influences the consumer food waste behavior [70]. AOC has proven to be the predictor of recycling behavioral intention [23,71]. It has been discovered that when consumers are aware of the negative effects of waste on the environment and people, they are more likely to reuse, donate, or resell their waste materials [7]. Therefore, it can be hypothesized:

**Hypothesis 2a** **(H2a).**
*Awareness of consequences has a significant impact on consumer’s food waste reduction behavior.*


**Hypothesis 2b** **(H2b).***Awareness of consequences has a significant impact on consumer’s food waste reusing behavior*.

**Hypothesis 2c** **(H2c).***Awareness of consequences has a significant impact on consumer’s food waste recycling behavior*.

#### 2.2.4. Habit and Waste Reduction Behaviors

Habits are described as a “learned automatic response that maintains repetitive actions in certain situations” [72]. Habit (HAB) was considered as the main construction to influence e-waste to recycle behavior [73]; significantly influencing recycling intention towards e-waste [74]. Habit as a psychological factor that influences intention [75] has a considerable impact on behavior [76], predicts behavior [77], and also influences pro-environmental behavior, [68]. In addition, there is a positive influence on both the intention and behavior to reduce waste [71] and it has been found to be a significant predictor of food waste reduction behavior [50,72,78]. HAB was a significant construct to food waste behavior [32]. Therefore, based on the above literature it can be proposed:

**Hypothesis 3a** **(H3a).***Habit has a significant impact on consumer’s food waste reduction behavior*.

**Hypothesis 3b** **(H3b).***Habit has a significant impact on consumer’s food waste reusing behavior*.

**Hypothesis 3c** **(H3c).***Habit has a significant impact on consumer’s food waste recycling behavior*.

#### 2.2.5. Financial Concern and Waste Reduction Behaviors

Consumers considered financial concern (FCR) as an important factor that contributed towards reducing food waste. Food waste was linked to financial waste and consumers displayed their concern to reduce it [79,80,81]. However, consumers also took it as an obstacle that prevents them reducing waste, as the cost to recycle food waste is more than composting [82]. Consumers described FCR as important in food waste [6]. The experimental result revealed an association between financial concern and food waste reduction behavior in both control and treatment household groups [83]. Consumers expressed more concern about financial loss as they waste food but results found that financial concern did not have much effect on food waste reduction [84]. Food waste was described by consumers as a waste of money, and indicating emotion of guilt while showing the behavior of wasting food [85] and there was found a strong negative association with food waste [21,86]. Young consumers exhibited more concern towards financial factors associated with food waste as compared to the rest. Previous literature revealed financial concerns in wasting food and predicted food waste reduction intention [80]. Consumer food choice, financial attitude, and social and situational factors are important to describe food waste behavior [87]. Therefore, it can be hypothesized that:

**Hypothesis 4a** **(H4a).***Financial concern has a significant impact on consumer’s food waste reduction behavior*.

**Hypothesis 4b** **(H4b).***Financial concern has a significant impact on consumer’s food waste reusing behavior*.

**Hypothesis 4c** **(H4c).***Financial concern has a significant impact on consumer’s food waste recycling behavior*.

#### 2.2.6. Research Model

The following diagram depicts the study’s conceptual context. The arrows represent all hypotheses (Figure 1).

## 3. Data and Methodology

### 3.1. Data Collection

The data used in this research study were gathered from a sample of young Pakistani consumers. The age of the individual was used as the primary criterion for sample inclusion. The respondents were all between the ages of 18 and 32. Among probability and non-probability sampling approaches [88], the current study used a convenience sampling technique. The key argument for choosing convenience sampling is that there is no way to compile a fixed list of potential young food waste consumers. As a result, probability sampling is not a practical method. Furthermore, convenience sampling may aid in the collection of replies from truly interested respondents while avoiding non-serious respondents who may have an impact on the research’s outcome [89]. Furthermore, non-probability sampling approaches were used in the majority of the research works from the food sector [8,90,91].

From September to November 2020, data were collected using a standardized questionnaire survey distributed through computer-assisted web interviewing (CAWI). The online survey was developed with Google Forms and distributed via email and the popular social network Facebook. Since the study focuses on younger consumers, the computer-assisted web interviewing (CAWI) methodology has been taken and literature has found younger people prefer to be part of the CAWI study [92]. Confidential reporting on the computer can be more honest and accurate than reports delivered by an interviewer orally [93]. Various studies, such as those conducted by [94,95], have examined the idea that computer self-completion provides more honest reports of embarrassing behaviors. A two-section online questionnaire was asked to be completed by participants. The first part gathered the social demographic features of participants (i.e., gender, age, and education). The second part of the survey focused on the attitudes and practices of the participants in relation to reducing food waste. During the online procedure, 492 complete and usable questionnaires were collected.

### 3.2. Measurement Scale

In order to obtain data from target respondents, we use a survey-based structured questionnaire. All of the research constructs have been adapted and conceptualized on the basis of previously established scales. In the first part of the questionnaire consumer’s demographic information was measured. All study variables based on the theoretical framework were measured in the second part of the survey. The measurement items for the construct were based on measurement scales from a variety of sources, and a five-point Likert scale (between strongly agree and strongly disagree) was used to assess the construct. One of the survey instruments developed was examined by a team of professionals (three researchers and one specialist) with expertise in food waste research for its validity [96]. Gao et al. [60] instrument that contains six items measured the awareness of consequences concepts (i.e., “In my opinion, recycling plastic waste is a major way to reduce pollution”). Issock et al. [97] instrument that contains four items measured the anticipated negative emotions (i.e., I feel guilty if not to reduce/reuse/recycle food waste”). Issock et al. [97] instrument that contains six items measured habits (i.e., “I separate my household waste materials automatically before the municipality collects it”). Visschers et al. [84] instrument that contains four items measured the financial concerns (i.e., “I cannot afford to pay for foods that are then discarded”). Barr et al. [98] instrument contains three items that measure the consumer’s food waste reduction behavior of young consumers (i.e., ‘In the next few weeks, I plan to reduce my food waste with more attention to buying’). Barr et al. [98] instrument was that contains five items measured the consumer’s food waste reusing behavior of young consumers (i.e., ‘I reuse leftover food because it can significantly benefit the environment’). Vinzi et al. [99] instrument contains five items that measure the consumer’s food waste recycling behavior of young consumers (i.e., ‘I resell much of my leftover food for economic reasons’).

### 3.3. Data Processing Procedure

To examine the hypothesized model, the Partial Least Square–Structural Equation Modelling (PLS-SEM) was used. This strategy is designed to develop and investigate complex relations among factors or multiple constructs. In addition, this strategy is helpful in the analysis of hypotheses and the correlation between various variables. It also helps to test the cause and effect relation among latent factors. PLS-SEM is a model of two phases. The first model is referred to as a ‘measurement model,’ while the following model is referred to as a ‘structural model.’ These two models validate the research model with different techniques [99].

## 4. Data Analysis

### 4.1. Respondent Profile Analysis

The respondent’s profiles are presented in this part of the research. Thus, the gender, age, and education of the 492 persons are considered. The number of males is 222 and the number of females is 270. There are 253 people between the ages of 18 and 22. In total, 151 people are between the ages of 23 and 27 and 69 people are between the ages of 28 and 32. Most of the people have bachelor-level education, i.e., 229. Results are illustrated in Table 1.

### 4.2. Measurement Model

The current study applied the PLS (partial-least-squares) modeling technique for structural equations. The outer loads are initially analyzed in the measurement model. By verifying the observed constructs and their respective items, the significance of the measurement model is improved. For this purpose, the outer loading of each variable item is inspected. Any item with an outer loading below 0.50 is excluded based on the criterion. The awareness of consequences is measured through five items (i.e., AOC1, AOC 2, AOC3, AOC4, and AOC5). Anticipated negative emotions are measured through four items (i.e., ANE1, ANE2, ANE3, and ANE4) and one item was excluded due to poor outer loading. Habit is measured through six items (i.e., HAB1, HAB2, HAB3, HAB4, HAB5, and HAB6) and no item was excluded. Financial concern is measured through four items (i.e., FCR1, FCR2, FCR3, and FCR4) and no item was excluded. Reduce food waste is measured through five items (i.e., RED1, RED2, RED3, RED4, and RED5) and two items were excluded. Reuse food waste is measured through five items (i.e., REU1, REU2, REU3, REU4, and REU5) and no item was excluded. Recycle food waste is measured through five items (i.e., REC1, REC2, REC3, REC4, and REC5) and no item was excluded (For the statement of each item, see Appendix A (Table A1). Outer loading results were shown in Table 2.

The next part is to test the reliability and validity analysis of all constructs after a depth analysis of external loadings. Composite reliability and Cronbach alpha are two main criteria used to analyze reliability (internal consistency). The first measure used was Cronbach alpha. It provides the reliability assessment while using correlation among variables if equality of reliability in all variables is maintained. Cronbach alpha’s final values ranged to 0.72–0.89 with a high level of reliability by [100], i.e., >0.70. Table 2 showed the results. Composite reliability is the second indicator for internal consistency. The composite reliability was tested using outer loadings of all variables. Composite reliability final values ranged from 0.88–0.92 was demonstrated to have high reliability i.e., >0.70 [101]. Table 2 showed the results. Convergent validity can be used to test the correlation between all items in the respective variable. The Average Variance Extracted (AVE) is evaluated for convergent validity testing of variables. The results showed that variables ranked between 0.59 and 0.71 for convergent validity. All constructs demonstrated high convergent validity because AVE is higher than thresholds 0.50, as suggested [100]. Table 2 presents the results of convergent validity. Finally, all research variables’ discriminant validity is assessed. The Heterotrait-Monotrait (HTMT) approach of assessing discriminant validity is used for this purpose. According to the HTMT method, the HTMT value should be less than 0.85. Table 3 summarizes the results that demonstrated discriminant validity exists.

### 4.3. Structural Model

Following the analysis of the measurement model, the structural model was examined for research hypotheses across all constructs (see Figure 2). Hair et al. [102] suggested five measures for evaluating the structural model: multi-collinearity evaluation, the significance of hypotheses, evaluation of R^2^, assessment of f^2^, and Q^2^. The variance inflation factor test was used for all constructs as a first step in determining multicollinearity. The results showed that there was no multicollinearity problem because the VIF score was less than 3.3, which is the recommended threshold by [102]. The findings are summarized in Table 4 below.

Hypotheses are checked in the second phase. The initial hypothesis i.e., H1a, value exhibited that β = 0.44, *p* < 0.00 which revealed that awareness of consequences (AOC) has a significant positive impact on reducing food waste behavior. In H1b, the value exhibited that β = 0.18, *p* < 0.00 which revealed that AOC has a significant positive impact on reusing food waste behavior. In H1c, the value exhibited that β = 0.44, *p* < 0.00 which revealed that AOC has a significant positive impact on recycling food waste behavior. In the second hypothesis, H2a, the value exhibited that β = 0.16, *p* < 0.00 which revealed that anticipated negative emotions (ANE) have a significant positive impact on reducing food waste behavior. In H2b, the value exhibited that β = 0.35, *p* < 0.00 which revealed that ANE has a significant positive impact on reusing food waste behavior. In H2c, the value exhibited that β = 0.29, *p* < 0.00 which revealed that ANE has a significant positive impact on recycling food waste behavior. In the third hypothesis H3a, the value exhibited that β = 0.13, *p* < 0.00 which revealed that habit (HAB) has a significant positive impact on reducing food waste behavior. In H3b, the value exhibited that β = 0.17, *p* < 0.00 which revealed that HAB has a significant positive impact on reusing food waste behavior. In H3c, the value exhibited that β = 0.16, *p* < 0.00 which revealed that HAB has a significant positive impact on recycling food waste behavior (detail summary is provided in Table 5).

Finally, in the fourth hypothesis H4a, the value exhibited that β = 0.09, *p* < 0.00 which revealed that financial concerns (FCR) have a significant positive impact on reducing food waste behavior. In H4b, the value exhibited that β = 0.23, *p* < 0.00 which revealed that FCR has a significant positive impact on reusing food waste behavior. In the last hypothesis H4c, value exhibited that β = 0.24, *p* < 0.00 which revealed that FCR has a significant positive impact on recycling food waste behavior. In the next step, R^2^ (determination coefficient) measures the model’s prediction accuracy. Further, effect size (f^2^) is measured. The result exposed that awareness of consequences has an f^2^ value of 0.20 with reducing food waste behavior which illustrates moderate effect size (i.e., value 0.15 to 0.35). Anticipated negative emotions have an f^2^ value of 0.16 with reusing food waste behavior that illustrates moderate effect size (i.e., value 0.15 to 0.35), whereas other constructs relationships have weak effect size (<0.15) as recommended by [102]. Results are exhibited in Table 6.

In the last step, a blindfolding technique measures predictive relevance (i.e., Q^2^). Results disclosed that Q^2^ values for reducing food waste behavior, reusing food waste behavior, and recycling food waste behavior are 0.28, 0.32, and 0.34, respectively, which are above the criterion (greater than 0) according to [102]. Results are exhibited in Table 7.

## 5. Discussion and Concluding Remarks

The appropriate way to reduce food waste is through reducing, reusing, and recycling. This study is an initiative to scrutinize the food waste reduction behavior of the consumer. Since a major part of food is wasted in the last part of the food supply chain, this study is an attempt to analyze various factors that can reduce food waste at the consumer level. Globally, food waste is criticized and encouraged for reduction. In spite of food waste being recognized as something to be reduced [80,103], consumer food waste remains an important concern that has an adverse effect on society. This study examined waste reduction behavior in terms of 3Rs (reduce, reuse, and recycle food waste).

This study addresses a major gap in past food studies, a lack of young consumer’s awareness about the consequences of food waste, the habit of wasting, not feeling guilt about wasted food, and a lack of concern about wasting money as they waste food. In the developed world, consumers actively participate in recycling and reducing food waste. On the other hand, in Pakistan, the reduction, reusing, and recycling of food waste are not considered an issue. The awareness of consequences is an important factor that supports reducing, reusing, and recycling food waste. Consumers, being aware of the consequences of food waste on the environment and society, prefer to reduce food waste and feel guilty if not reducing waste. Contrarily, consumers are continuing to wasting food on daily basis. The situation of food waste in Pakistan is not very good, where a large population has not enough food to eat. Consumers are not aware of financial loss as they waste food and leave food on their plates. As they are unaware of the consequences of food waste they do not reuse their leftover food and vice versa [104].

The theory of interpersonal behavior was used to bridge the gap observed in the Theory of Planned Behavior (TPB). TPB uses a cognitive approach whereas the TIB also focused on consumer habits and emotions. It integrates and examines individuals’ habits, negative emotions, awareness of consequences, and financial concerns. The effects of these independent variables were considered as the predictors of food waste reduction, reusing, and reducing behavior.

Results showed consumers reduced and recycled food waste rather than reuse it. Moreover, study results revealed the significance of habit and anticipated emotion of guilt to contributing towards food waste reduction behavior (reduction, reusing, and recycling), similar to previous findings. Studies expect that wasting food provokes a negative feeling of guilt but results showed insignificant influence on behaviors of food waste reduction, reusing, and recycling. Study results contradict the previous results where negative emotion strongly influences their behavior, when they feel guilty to show appropriate behavior and recognized their behavior causing damage to the environment [58]. The lack of emotion of guilt among consumers evokes guilt to encourage the behavior to reduce, reuse, and recycle food waste [105]. The result indicated that negative emotions are unrelated but it does not mean that factor is unimportant [106]. Consumers who have less concern with money consume more, order more, and put more on the plate, resulting in more food waste due to habit and no feeling of guilt [45]. The role of habit in food waste reduction, reusing, and recycling behaviors is important. Habit has a fundamental role in acquiring environmentally responsible behavior [68]. However, food waste reduction behavior (3R) that is repeated frequently influences habits to develop [97]. Habits to reduce, reuse and recycle food waste are considered as pro-environmental behavior and once a consumer engages in the habit of reducing food waste they are more likely to practice food waste reduction everywhere, whether eating at home, at restaurants, or elsewhere. Moreover, the habit of reducing food waste has financial benefits too. In developing countries such as Pakistan, habits can save money and reduce food waste [107]. Financial concern is related to food waste reduction behavior, as found in past studies [21,79]. Consumers showed different perspectives related to food waste and were more concerned with the loss of money as they wasted food. On the other, hand recycling and reduction of food waste are more related to saving money as buying less can reduce waste and recycling can minimize the environmental cost [86]. In Pakistan, where money has more value, food waste habits still persist. Consumers are more conscious of the cost of food at home and restaurants, but those who show financial concern produce less waste but evidence opposing it [24]. Financial concern is more important to reducing food waste and changing lifestyles [108]. However, financial concern for food waste recycling is considered a barrier due to the cost of recycling.

There are several useful implications to apply. In Pakistan, there is a lack of legal and regulatory binding regarding food waste reduction management and ways to handle it. Second, consumers need to be educated about the consequences of food waste through promotional campaigns and programs. Promotion about price and quantity of food to reduce food left on the plate may enhance the awareness of consumers. Consumer awareness about the positive and negative consequences of food waste reduction through information on media as well as its impact on individuals and society is important. Thirdly, the young generation should be educated through social media about food waste reduction and should be kept informed about the consequences too. In order to create awareness among young consumers, approaches such as celebrity endorsement of reducing food waste might be used. Moreover, advertisements on popular media such as social media will encourage consumers to reduce food waste and share it with their community groups to enhance their knowledge about the food waste issue.

For the practical implications of managing consumer food waste at restaurants and hotels, first it is important to inform consumers about the sufficient order of one person and suggest options to pack their leftovers to take them home to reuse them. Secondly, advertise food waste as money waste. Thirdly, promote social media campaigns about the restaurant/hotel’s contributions to food waste reduction to create awareness, emotions, and change habits. Theoretically, an important contribution of the study was in examining the direct link of awareness of consequences, habit, emotion, and financial concern on the behavior of food waste reduction (reduction, reusing & recycling). The importance of cognitive factors of emotions and habit in describing food waste reduction, reusing, and recycling behavior indicated that selecting habit and emotion will be a useful strategy to change behavior. Our results suggested that factors have important contributions to find out complex food waste reduction behavior and ignoring cognitive factors of habit and emotion will make it impossible to understand the behavior fully. Studies have exhibited that all factors, including awareness of consequences, habit, anticipated emotion of guilt, and financial concern, have a considerable influence on food waste reduction, reuse, and recycling behavior. Direct relationships with the behavior were examined and all hypotheses were confirmed to predict the behaviors. Our result highlighted the significance of non-cognitive factors of emotion and habit to food waste behaviors (reduction, reuse, and recycling) in order to change restaurants’ consumer behavior. These findings are important to practitioners and researchers to improve waste reduction behavior by engaging emotion through media and also develop and change the habit of food waste in young consumers through social media.

There are some limitations to the study. First, the sample was selected from two cities in the country. Second, only consumer perspective was measured and restaurants/hotel employees and owner perspectives need to be investigated. Thirdly, we examine the observed food waste reduction behavior depending on to recall their memory when eating out. Fourthly, we only considered the negative emotion of guilt to reduce food waste and ignored the positive emotion. The current research was related to consumers who were eating out, but consumers at households who order food online in current online trend & in COVID scenario, and consumers at institutions such as offices, hospitals, and educational institutions can be examined for future research. Moreover, group comparison between income level and urban vs rural consumers will give useful insight into how food waste is perceived. The current study considered only food waste, and a future study can examine the other sources of consumer waste such as plastic packaging and expiry dates on food items.

## Figures and Tables

**Figure 1 ijerph-18-12457-f001:**
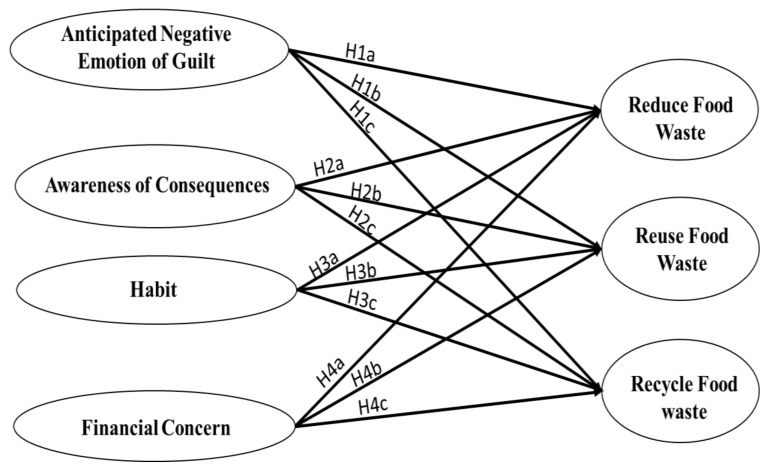
Conceptual Framework.

**Figure 2 ijerph-18-12457-f002:**
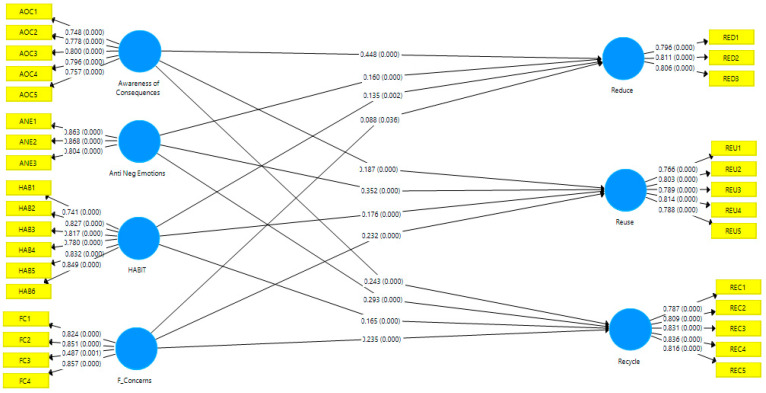
Structural Model.

**Table 1 ijerph-18-12457-t001:** Sample Characteristics (*N* = 492).

Demographic	Category	Percentage (Freq)
Gender	Male	45.2 (222)
	Female	54.8 (270)
Age (In years)	Less than 18	3.9 (19)
	18–22 years	51.4 (253)
	23–27 years	30.7 (151)
	28–32 years	14.0 (69)
Education	High School	21.6 (106)
	Professional degree/vocational school	6.7 (33)
	Bachelors	46.5 (229)
	Masters	22.1 (109)
	Doctorate	3.1 (15)

**Table 2 ijerph-18-12457-t002:** Result of Measurement Model.

Constructs	Code	Outer Loadings	Cronbach’s Alpha	Composite Reliability	Average Variance Extracted
Awareness of Consequences (AOC)	AOC1	0.75	0.84	0.88	0.60
	AOC2	0.78			
	AOC3	0.80			
	AOC4	0.80			
	AOC5	0.76			
Anticipated Negative Emotions (ANE)	ANE1	0.86	0.80	0.88	0.71
	ANE2	0.87			
	ANE3	0.80			
	ANE4	0.45			
	ANE5	0.41			
Habit (HAB)	HAB1	0.74	0.89	0.92	0.65
	HAB2	0.83			
	HAB3	0.82			
	HAB4	0.78			
	HAB5	0.83			
	HAB6	0.85			
Financial Concerns (FCR)	FCR1	0.82	0.81	0.89	0.73
	FCR2	0.85			
	FCR3	0.49			
	FCR4	0.86			
Reduce Food Waste Behavior (RED)	RED1	0.80	0.72	0.85	0.65
	RED2	0.81			
	RED3	0.81			
	RED4	0.48			
	RED5	0.41			
Reuse Food Waste Behavior (REU)	REU1	0.77	0.85	0.89	0.63
	REU2	0.80			
	REU3	0.79			
	REU4	0.81			
	REU5	0.79			
Recycle Food Waste Behavior (REC)	REC1	0.79	0.87	0.91	0.67
	REC2	0.81			
	REC3	0.83			
	REC4	0.84			
	REC5	0.82			

**Table 3 ijerph-18-12457-t003:** Discriminant Validity HTMT Ratio.

	AOC	ANE	HAB	FCR	RED	REU	REC
Awareness of consequences							
Anticipated negative emotions	0.73						
Habit	0.41	0.37					
Financial concerns	0.61	0.46	0.61				
Reduce food waste	0.82	0.66	0.49	0.58			
Reuse food waste	0.68	0.73	0.55	0.66	0.57		
Recycle food waste	0.70	0.69	0.53	0.66	0.65	0.84	

AOC stands for ‘Awareness of consequences’, ANE = Anticipated negative emotions. HAB = Habit, FCR = Financial concerns, RED = Reduce food waste, REU=Reuse food waste, REC = Recycle food waste.

**Table 4 ijerph-18-12457-t004:** Collinearity Test.

	Reduce	Reuse	Recycle
Awareness of consequences	1.86	1.86	1.86
Anticipated negative emotions	1.62	1.62	1.62
Habit	1.41	1.41	1.41
Financial concerns	1.65	1.65	1.65

**Table 5 ijerph-18-12457-t005:** Analysis of Hypotheses.

Structural Paths	β	*p*-Value	Results
H1a: Awareness of consequences → Reducing Behavior	0.44	0.00	Supported
H1b: Awareness of consequences → Reusing Behavior	0.18	0.00	Supported
H1c: Awareness of consequences → Recycling Behavior	0.23	0.00	Supported
H2a: Anticipated negative emotions → Reducing Behavior	0.16	0.00	Supported
H2b: Anticipated negative emotions → Reusing Behavior	0.35	0.00	Supported
H2c: Anticipated negative emotions → Recycling Behavior	0.29	0.00	Supported
H3a: Habit → Reducing Behavior	0.13	0.00	Supported
H3b: Habit → Reusing Behavior	0.17	0.00	Supported
H3c: Habit → Recycling Behavior	0.16	0.00	Supported
H4a: Financial concerns → Reducing Behavior	0.09	0.00	Supported
H4b: Financial concerns → Reusing Behavior	0.23	0.00	Supported
H4c: Financial concerns → Recycling Behavior	0.24	0.00	Supported

**Table 6 ijerph-18-12457-t006:** Effect Size (f^2^).

	Reduce	Reuse	Recycle
Awareness of consequences	0.20	0.04	0.06
Anticipated negative emotions	0.03	0.16	0.11
Habit	0.02	0.05	0.04
Financial concerns	0.01	0.07	0.08

**Table 7 ijerph-18-12457-t007:** Analysis of R^2^ and Q^2^.

	R^2^	Q^2^
Reducing food waste behavior	0.46	0.28
Reusing food waste behavior	0.53	0.32
Recycling food waste behavior	0.52	0.34

## Data Availability

Data are contained within the article.

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
