# Peer review of "Antecedents of Consumer Food Waste Reduction Behavior: Psychological and Financial Concerns through the Lens of the Theory of Interpersonal Behavior"

_ijerph, 2021, doi:10.3390/ijerph182312457_

Round 1

Reviewer 1 Report

Dear Authors,

Thank you for your effort, and please find my comments related to your manuscript. 

The presented paper deals with an important topic. As the Authors highlighted in the Introduction “The high quantity of restaurant food waste is a major challenge to the food service sector and concern in developing countries where eating out of home is an emerging trend. In these markets, a large portion of restaurant food waste comes from consumer’s plates, which emphasizes the changes in consumer’s behavior to reduce food waste”.

Additionally, I believe the set of data collected by the Authors allows for complex analysis and important findings.

The strengths of the paper are as follows:

- important topic

- quantitative approach

However, in my opinion, the whole paper needs some rework to be more clear and to use the whole potential of the collected data. I will present my remarks in two sections: “general” and “details”.

General remark 1 – the aim of the study

The aim of the study in presented only in the abstract in the following way: “The study aimed to understand the factors that affect food waste reduction, reuse, and recycling behavior of restaurants’ consumers”.

The aim of the paper should be presented not only in the abstract and expressed clearer as the word “understand” seems to be too general.  

General remark 2 – methodology and results presentation

Due to the fact that the methodology description is not detailed enough, the result presentation is unclear. The Authors stated that for the model their used 7 constructs (from “Awareness of Consequences” to “Recycle Food Waste Behavior”, for each construct from 4 to 6 items were used. However, it is not presented what items actually have been used. This makes the whole paper difficult to understand and rather general. For instance, the Authors stated that (line 402) “that FCR has a significant positive impact on reusing food waste behaviour”. This is an important statement, but it is not possible to understand what it really means, what items create “Financial Concerns” and what reusing behaviour did Authors mean. In my opinion, you did not use the whole potential of the data you collected.

Therefore I would recommend:

  1. To extend the methodological part and upload a questionnaire to the paper as an appendix…
  2. …then, to extend results description and conclusions accordingly.

Detailed remarks

Lines 14 – 27

The whole abstract need to be restructured – there is now to many repetitions, e.g. lines 18-19 vs lines 22-23.

Lines 31 – 70

Introducing the problem it would be good to supplement literature with: https://doi.org/10.3390/agriculture11010019; https://doi.org/10.3390/agriculture11100936

Lines 118 – 128

Section 1.1.2. Practical Significance – in fact, this part does not present practical implication but introduces how the paper and research is structured => I would suggest moving that part to the introduction. “Practical Significance” should be extended or you can present it together with “Theoretical Significance”

Line 134

You have stated “Due to the complex human behavior, the most common TPB seems to be unable to…” => TPB did not appear it the text earlier, please use the theory of planned behaviour, then continue with the abbreviation.

Lines 274-275

The sentence “Detailed information about the data collection process, study sample, measurement scales, and data analysis methodology are provided in this section”, is not needed, I suggest to delete.

Lines 274-275:

“492 usable questionnaires were produced during the online procedure”. It should be “492 interviews were conducted”, moreover – please do not start the sentence with a number.

Lines 279-280

 “The non-probability sampling i.e. convenience-sampling technique was used in this study” => how did you control a sample structure, e.g. dose you sample reflects the distribution of the gender in total population

Line 311

3.3. Methodology => I think this title is not appropriate. It is rather “statistical procedure” or “data processing procedure”.

Line 419

Figure 2. Structural Model => is of a very low quality.

Good luck

Author Response

Thank you very much for your invaluable comments and suggestions, which have improved the revised version significantly.

We would also like to send our appreciation to you for your time and efforts in reviewing our paper. We would like to thank you for your following comments:

  • Thank you for your effort
  • The presented paper deals with an important topic.
  • As the Authors highlighted in the Introduction “The high quantity of restaurant food waste is a major challenge to the food service sector and concern in developing countries where eating out of home is an emerging trend.
  • In these markets, a large portion of restaurant food waste comes from consumer’s plates, which emphasizes the changes in consumer’s behavior to reduce food waste”.
  • Additionally, I believe the set of data collected by the Authors allows for complex analysis and important findings.
  • The strengths of the paper are as follows: important topic, quantitative approach

We would also like to send our appreciation to you for your time and efforts in reviewing our paper and for providing the excellent comments. Below are our responses to your helpful comments and suggestions.

Question 1. Does the introduction provide sufficient background and include all relevant references?        (must be improved)

Answer 1:  Thank you very much for your advice. We have improved the introduction to provide sufficient background and include all relevant references in our revised manuscript.

Question 2. Is the research design appropriate? (can be improved)

Answer 2:  Thank you very much for your advice. We have improved the research design and made it more appropriate in our revised manuscript.

Question 3. Are the methods adequately described?        (must be improved)

Answer 3:  Thank you very much for your advice. We have described the methods adequately in our revised manuscript.

Question 4. Are the results clearly presented? (must be improved)

Answer 4:  Thank you very much for your advice. We have improved the presentation of our result to make it clearer in our revised manuscript.

Question 5: Are the conclusions supported by the results? (must be improved)

Answer 5:  Thank you very much for your advice. We have improved the conclusions which are supported by the results in our revised manuscript.

Question 6. The aim of the study in presented only in the abstract in the following way: “The study aimed to understand the factors that affect food waste reduction, reuse, and recycling behavior of restaurants’ consumers”.  The aim of the paper should be presented not only in the abstract and expressed clearer as the word “understand” seems to be too general. 

Answer 6:  Thank you very much for your advice. We have presented the abstract and expressed clearer in our revised manuscript.

Question 7. Due to the fact that the methodology description is not detailed enough, the result presentation is unclear. The Authors stated that for the model their used 7 constructs (from “Awareness of Consequences” to “Recycle Food Waste Behavior”, for each construct from 4 to 6 items were used. However, it is not presented what items actually have been used. This makes the whole paper difficult to understand and rather general. For instance, the Authors stated that (line 402) “that FCR has a significant positive impact on reusing food waste behaviour”. This is an important statement, but it is not possible to understand what it really means, what items create “Financial Concerns” and what reusing behaviour did Authors mean. In my opinion, you did not use the whole potential of the data you collected.

Therefore I would recommend:

  1. To extend the methodological part and upload a questionnaire to the paper as an appendix…
  2. …then, to extend results description and conclusions accordingly.

Answer 7:  Thank you very much for your advice. We have extended the methodological part and uploaded a questionnaire to the paper as an appendix, and extended results description and conclusions accordingly in our revised manuscript.

Question 8. The whole abstract need to be restructured – there is now to many repetitions, e.g. lines 18-19 vs lines 22-23.

 Answer 8:  Thank you very much for your advice. We have restricted the whole abstract and reduced any repetitions in our revised manuscript.

Question 9. Introducing the problem it would be good to supplement literature with: https://doi.org/10.3390/agriculture11010019; https://doi.org/10.3390/agriculture11100936

Answer 9:  Thank you very much for your advice and information. We have cited your recommended papers and discuss the issue in our revised manuscript.

Question 10. Section 1.1.2. Practical Significance – in fact, this part does not present practical implication but introduces how the paper and research is structured => I would suggest moving that part to the introduction. “Practical Significance” should be extended or you can present it together with “Theoretical Significance”

Answer 10:  Thank you very much for your advice. We have moved it to the introduction and extended “Practical Significance” and presented it together with “Theoretical Significance” in our revised manuscript.

Question 11. You have stated “Due to the complex human behavior, the most common TPB seems to be unable to…” => TPB did not appear it the text earlier, please use the theory of planned behaviour, then continue with the abbreviation.

Answer 11:  Thank you very much for your advice. We have used the theory of planned behaviour, and then continued with the abbreviation in our revised manuscript.

Question 12. The sentence “Detailed information about the data collection process, study sample, measurement scales, and data analysis methodology are provided in this section”, is not needed, I suggest to delete.

Answer 12:  Thank you very much for your advice. We have deleted the sentence in our revised manuscript.

Question 13. “492 usable questionnaires were produced during the online procedure”. It should be “492 interviews were conducted”, moreover – please do not start the sentence with a number.

Answer 13:  Thank you very much for your advice and suggestion. We have addressed the issue in our revised manuscript.

Question 14. “The non-probability sampling i.e. convenience-sampling technique was used in this study” => how did you control a sample structure, e.g. dose you sample reflects the distribution of the gender in total population

Answer 14:  Thank you very much for your advice. We have discussed our sampling approach clearly in our revised manuscript.

Question 15. 3.3. Methodology => I think this title is not appropriate. It is rather “statistical procedure” or “data processing procedure”.

Answer 15:  Thank you very much for your advice. We have changed the title of Section 3.3 to be “data processing and statistical procedures” in our revised manuscript.

Question 16. Figure 2. Structural Model => is of a very low quality.

Answer 16:  Thank you very much for your advice. We have addressed the issue in our revised manuscript.

We hope that you will find this manuscript suitable to be included in an upcoming issue of your publication.

Reviewer 2 Report

The paper brings into attention a topic of interest regarding the consumer food waste behavior, the authors proposing and testing a model aimed to explain variables such as: reduce food waste, reuse food waste and recycle food waste.

The title should not include an abbreviation, even though it was explained  into the text. 

The research methodology should mention the items used to define each latent variable alongside with the studies taken into account to propose these items, for each of the seven latent variables.

Author Response

Thank you very much for your invaluable comments and suggestions, which have improved the revised version significantly.

We would also like to send our appreciation to you for your time and efforts in reviewing our paper. We would like to thank you for your following comments:

  • Is the research design appropriate? (yes)  
  • Are the conclusions supported by the results? (yes)
  • The paper brings into attention a topic of interest regarding the consumer food waste behavior,
  • the authors proposing and testing a model aimed to explain variables such as: reduce food waste, reuse food waste and recycle food waste.

We would also like to send our appreciation to you for your time and efforts in reviewing our paper and for providing the excellent comments. Below are our responses to your helpful comments and suggestions.

Question 1. Does the introduction provide sufficient background and include all relevant references?        (can be improved)

Answer 1:  Thank you very much for your advice. We have improved the introduction to provide sufficient background and include all relevant references in our revised manuscript.

Question 2. Are the methods adequately described?        (must be improved)

Answer 2:  Thank you very much for your advice. We have described the methods adequately in our revised manuscript.

Question 3. Are the results clearly presented? (can be improved)

Answer 3:  Thank you very much for your advice. We have improved the presentation of our result to make it clearer in our revised manuscript.

Question 4: Moderate English changes required

Answer 4:  Thank you very much for your advice. We have read and polished our paper carefully.

Question 5. The title should not include an abbreviation, even though it was explained  into the text.

Answer 5:  Thank you very much for your advice. We have addressed the issue in our revised manuscript.

Question 6. The research methodology should mention the items used to define each latent variable alongside with the studies taken into account to propose these items, for each of the seven latent variables.

Answer 6:  Thank you very much for your advice. We have mentioned the items used to define each latent variable alongside with the studies taken into account to propose these items, for each of the seven latent variables in our revised manuscript.

We hope that you will find this manuscript suitable to be included in an upcoming issue of your publication.

Round 2

Reviewer 1 Report

Dear Authors,

Thank you very much for updated paper and please find my comments. I find this version more clear to the readier. The most important changes are related to the introduction and abstract (updated version is much better). More detail description of the methodological part allow to understand better how latent variables were constructed. 
I would only suggest to check the language correctness.

Good luck